# Targeting Hedgehog Signalling through the Ubiquitylation Process: The Multiple Roles of the HECT-E3 Ligase Itch

**DOI:** 10.3390/cells8020098

**Published:** 2019-01-29

**Authors:** Paola Infante, Ludovica Lospinoso Severini, Flavia Bernardi, Francesca Bufalieri, Lucia Di Marcotullio

**Affiliations:** 1Center for Life NanoScience@Sapienza, Istituto Italiano di Tecnologia, 00161 Rome, Italy; paola.infante@iit.it; 2Department of Molecular Medicine, University of Rome La Sapienza, 00161 Rome, Italy; ludovica.lospinososeverini@uniroma1.it (L.L.S.); flavia.bernardi@uniroma1.it (F.B.); francesca.bufalieri@uniroma1.it (F.B.); 3Istituto Pasteur-Fondazione Cenci Bolognetti, University of Rome La Sapienza, 00161 Rome, Italy

**Keywords:** Hedgehog, cancer, ubiquitylation, Itch, Numb, β-arrestin2, GLI1, SuFu, Patched1

## Abstract

Hedgehog signalling (Hh) is a developmental conserved pathway strongly involved in cancers when deregulated. This important pathway is orchestrated by numerous regulators, transduces through distinct routes and is finely tuned at multiple levels. In this regard, ubiquitylation processes stand as essential for controlling Hh pathway output. Although this post-translational modification governs proteins turnover, it is also implicated in non-proteolytic events, thereby regulating the most important cellular functions. The HECT E3 ligase Itch, well known to control immune response, is emerging to have a pivotal role in tumorigenesis. By illustrating Itch specificities on Hh signalling key components, here we review the role of this HECT E3 ubiquitin ligase in suppressing Hh-dependent tumours and explore its potential as promising target for innovative therapeutic approaches.

## 1. Introduction

Hedgehog (Hh) signalling cascade is an evolutionarily conserved pathway that plays a pivotal role in developmental processes, tissues homeostasis and stem cell/progenitors maintenance [1,2]. Indeed, Hh signals act as long-range morphogens to control cell patterning and differentiation in several embryonic tissues, such as the central nervous system (CNS). Moreover, Hh functions also as a mitogen regulating both cell proliferation and differentiation of granule cells progenitors (GCPs) in cerebellum [3]. Subversion of these Hh-regulated events determine uncontrolled cell growth that may predispose to the onset of many tumours, including medulloblastoma (MB), the most common paediatric brain malignancy, particularly aggressive and characterized by poor prognosis [4,5]. 

In vertebrates, Hh pathway is initiated through the interaction of Hh ligands to the 12-transmembrane receptor Patched1 (PTCH1) that relieves its inhibitory effect on the co-receptor Smoothened (SMO), a 7-pass transmembrane protein. This interaction triggers the activation of the GLI zinc finger transcription factors that translocate in the nucleus and promote the transcription of several Hh target genes involved in the most important biological processes [6]. Three GLI family members have been identified in mammals: GLI1 functions exclusively as a transcriptional activator, whereas GLI3 and to less extend GLI2, mainly work as repressor [2]. A key event for GLIs activity is their dissociation from Suppressor of Fused (SuFu) protein, an important negative regulator of the pathway mutated in infant MB. 

Dysregulation of the Hh pathway observed in MB and other tumours can be caused by ligand-independent events, such as genetic mutations of pathway components (*PTCH*, *SMO*, *SuFu*, *REN*, *GLI1/GLI2* amplification) [7,8] or ligand-dependent mechanisms of activation [9,10]. Indeed, besides the canonical pathway, which involves the Hh-dependent activation of the GLI proteins [11], the existence of non-canonical routes further increases the complexity of the signalling. Two types of non-canonical pathways can be recognized: type I, which does not require SMO and type II, which shows SMO-dependent but GLI-independent mechanisms [12]. Given its complexity, the Hh pathway needs to be finely orchestrated in a synergic and dynamic way.

Nevertheless, molecular mechanisms underlying Hh signalling arrest are crucial for controlling the fates of the pathway after its activation. In this regard, ubiquitin-dependent proteolytic processes play a crucial role [13]. Ubiquitylation and degradation of GLI factors are the most well established mechanisms in limiting the signal duration and avoiding an uncontrolled cell proliferation that, otherwise, would lead towards malignancy [14,15,16,17]. In the last years, other key components of Hh pathway, such as SMO, PTCH and SuFu, are emerging as tightly regulated through ubiquitylation processes, thus indicating that this post-translational modification serves as a general mechanism in the dynamic regulation of the Hh pathway.

## 2. Ubiquitylation Processes: Not Only a Degradative Pathway

Ubiquitylation is an elaborated post-translational modification that regulates a wide spectrum of cellular functions including protein degradation, endocytosis, sorting and trafficking of transmembrane proteins. This process is governed by an enzymatic cascade involving the E1, E2 and E3 enzymes and results in the transferring of ubiquitin (Ub) molecules onto substrate proteins. First, the ubiquitin-activating enzyme E1 forms a thioester bond between its active cysteine and the carboxyl-terminal glycine-76 residue of the ubiquitin in an ATP-dependent manner; then, the activated ubiquitin is transferred to the E2 ubiquitin-conjugating enzyme. In this series of catalytic reactions, the E3 ubiquitin ligases (E3s) are the major determinants that recruit ubiquitin-E2 complexes facilitating the transfer of activated ubiquitin from E2 to target proteins and provide specificity for substrate recognition [18,19] (Figure 1). Based on their functional domains, E3s can be subdivided into three families: the Really Interesting New Gene (RING), the U-box and the Homologous to the E6-associated protein Carboxyl-Terminus (HECT) types. The latter possess an intrinsic catalytic activity that directly allows ubiquitin transfer to their target proteins.

Ubiquitin can be covalently linked to lysine residues of substrates or a preceding Ub moiety as poly-ubiquitin chains or by multi- or mono-ubiquitylation events. Each one of these mono-, multi- or poly-ubiquityla-tion processes results in a different fate of ubiquitylated target protein and functional outcome [20]. For example, protein poly-ubiquitylation frequently leads to the proteolytic processing of the target, allowing recognition by the 26S proteasome system.

The ubiquitylation process is commonly found in Hh signalling in which regulates the activity of major components of the transduction pathway such as SMO, PTCH, SuFu and GLI transcription factors. These events have the role of modulating the stability of proteins, resulting in degradation through the proteasomal or lysosomal machinery, with important consequences for the activation or inhibition of Hh signalling. 

Notably, in the absence of the Hh ligand, the generation of the transcriptional repressor form and the degradation of the activated form of GLI transcription factors represent an important nodal point in the Hh signalling. These events are governed by phosphorylation and multiple ubiquitin-dependent proteolytic processes involving two families of E3 ubiquitin ligase: the RING and the HECT E3s. In the absence of signal, PKA, GSK3β and CK1 mediate the phosphorylation on multiple sites of GLI proteins generating signals for the recruitment of the SCF^βTrCP^ ubiquitin ligase, a RING E3 ligase, which includes the F-box protein βTrCP, Skp1, Cullin1 and Rbx1. This event leads to the proteolytic processing of GLI3 and partly of GLI2, into repressor form and a complete proteasome-dependent degradation of GLI1 and GLI2 [15,21,22]. The degradation of GLI2 and GLI3 can also occur by SPOP/Cullin 3-based E3 ligase [23,24], whereas GLI1 can be also degraded by the Numb-activated Itch E3 ubiquitin ligase [25,26] or by PCAF upon genotoxic stress [27].

Although the ubiquitylation is a relevant mechanism to control protein degradation [28], it is also required for a variety of non-proteolytic functions, such as modulation of signal transduction, transcription, protein-protein interaction and DNA repair. The nature of ubiquitin attachment and the type of the linkage forming the poly-Ub chain determine the fate of the Ub-target protein [29]. Indeed, Ub chains can be built via one of seven lysine residues (K6, K11, K27, K31, K33, K48 and K63), all of which can form an iso-peptide linkage with another Ub molecule leading to the generation of different types of ubiquitin chain conformations each ones with different outcomes. Chains formed through the addition of ubiquitin exclusively by lysine 48 (K48) label proteins for degradation via the 26S proteasome and represent the well-studied and canonical function of protein ubiquitylation [28]. On the other hands, K63-linked ubiquitin chains typically act as non-degradative signal and are thought to positively regulate protein localization and protein complex formation [30,31,32]. In this regard, the control of GLI factors by SuFu protein is governed through a K63-linkage ubiquitylation of SuFu without affecting its stability [33].

## 3. The HECT E3 Ubiquitin Ligase Itch: An Overview

Itch or atrophin-1 interacting protein 4 (AIP4) and Suppressor of Deltex in *Drosophila*, was originally identified by genetic studies on murine coat colour alterations [34] and its deletion resulted in an itchy phenotype with constant skin scratching and multi-organ inflammation. The *Itch* gene encodes a protein of 864 amino acids with a relative molecular weight of 113 KDa expressed in various tissues, including spleen, brain, kidney and testis.

### 3.1. Structure of Itch

Itch is a monomeric HECT-type E3 ubiquitin ligase belonging to the neural precursor cell-expressed developmentally downregulated protein 4 (NEDD4) subgroup sharing a characteristic modular structure: a N-terminal lipid-interacting C2 domain, four WW domains for protein-protein interaction and a C-terminal catalytic HECT domain [35]. The C2 domain of Itch binds Ca^2+^ and phospholipids and works in targeting the E3 ligase to the intracellular membranes [36]. The interaction domain consists of four WW domains that mediate the association with substrates through a variety of proline-rich motifs, including PPxY, PPLP and proline-containing phosphoserine/phosphothreonine [26] (Figure 2A). The HECT domain possesses intrinsic enzymatic activity: it directly catalyses the ubiquitylation of the target protein by accepting ubiquitin molecule from an E2 enzyme. Then, the ubiquitin is transfer from HECT domain to the lysine residue of the substrate. HECT motif is composed of N-terminal and C-terminal halves separated by a flexible hinge region required for the E3 ligase activity. The N-terminal lobe contains an E2-ubiquitin binding site and the C-terminal lobe includes the active-site cysteine residue [37]. The disclosed HECT domain structure revealed the dynamic rearrangement of the two lobes, which are able to adopt a catalytically active “L-shaped” open conformation or an inactive “T-shaped” closed conformation [38]. The transition from the open to the close state of the HECT domain is expected to juxtapose the active sites of the E3 and the bound E2, although additional conformational change, probably around the hinge region, are necessary to allow the trans-thioesterification reaction. In addition, further movements of HECT domain may be required for the release of the E2 enzyme, the repositioning of ubiquitin substrates or for the formation of an ubiquitin chain [38,39]. As observed in several E3s NEDD4 family members, Itch is maintained in an inactive state by self-inhibition processes, suggesting that beyond HECT other its own domains act as regulators to prevent non-specific substrates ubiquitylation or self-ubiquitylation events. In this regard, crystallographic studies have recently highlighted that Itch WW2 domain and the linker region connecting WW2 and WW3 restrict the inter-domain mobility of the catalytic HECT domain, locking Itch in a closed inactive conformation [40].

### 3.2. Regulation of Itch

Itch plays a crucial role in the regulation of the immune response and inflammation signal [41,42]. Upon TCR stimulation, JNK promotes Itch activity and leads to ubiquitylation and degradation of JunB [41,43]. In addition, a growing number of evidence demonstrates a link between Itch and tumour development due to the broad spectrum of substrates controlled by its activity. In particular, transmembrane receptors, kinases and many transcriptional factors are targeted for mono-, multi- or poly-ubiquitylation by Itch, driving them to lysosomal or proteasomal degradation. Notably, non-proteolytic events have been also ascribed to the E3 activity of Itch [44,45,46,47] (Figure 2B); indeed, with few exceptions Itch catalyses the assembly of poly-ubiquitin chains through lysine 48-mediated linkages leading to proteasomal degradation of target proteins or lysine 63-linkage characterizing a regulatory pathway [33]. 

Itch activity is tightly regulated through various molecular mechanisms that mediate the switch between the active-open and inactive-closed conformation. In this regard, post-translational modifications or binding of adaptor proteins can alleviate the autoinhibitory intramolecular interactions and trigger the enzymatic activity. For example, Itch is subjected to regulation by phosphorylation that may modulate, both positively and negatively, its activity or the interaction with target proteins. Indeed, Itch is controlled by serine/threonine kinases such as ATM that phosphorylates Itch at serine-161 enhancing its activity and driving the ubiquitylation and degradation of c-FLIP-L and c-Jun [48] or by JNK1 [49], on three sites within Pro-rich region, leading to conformational changes that disrupt the self-inhibitory intramolecular interaction between the WWs and the HECT domains and augments the catalytic activity of Itch. Conversely, phosphorylation at tyrosine-371 within the Itch WW domain induced by Src kinase Fyn upon T-cell receptor (TCR) stimulation may alter the binding affinity between Itch and JunB and reduce JunB proteolysis [50]. 

In addition, similarly to other E3s, Itch autocatalytic activity negatively controls its own protein stability, although its degradation is prevented by ubiquitin-protease FAM/USP9X [51]. However, Itch autoubiquitylation can occur through the K63 linkage rather than K48 linkage, thus providing a non-proteolytic regulatory function and making Itch a relatively high stable protein under physiological condition [32].

Interactions of Itch with different accessory proteins, such as NEDD4 family-interacting proteins (NDFIPs), Numb and β-arrestins, have been described to modulate its substrate recruiting capacity, subcellular localization and enzymatic activity [26,33,52,53,54] (Figure 2B). Binding of NDFIP proteins induces the enzymatic activity of Itch releasing its autoinhibitory conformation and allowing the access of other PY-containing substrates. Indeed, NDFIP1 and NDFIP2 stimulates autoubiquitylation of Itch and subsequently enhance Itch-mediated ubiquitylation of JunB and endophilin, in particular NDFIP2 [54]. Itch adaptor proteins, such as β-arrestins, can be required for Itch activity to modulate cellular localization of the target proteins. Itch-dependent ubiquitylation of TRPV4 requires β-arrestin1 to promote internalization rather than degradation of TRPV4 [55]. The role of the Itch adaptor proteins Numb and β-arrestin2 and their regulation in the Hh pathway will be extensively discuss below.

## 4. Itch and Its Role in Canonical Hh Pathway

Itch activity is finely modulated by several mechanisms, whose alterations would be linked to tumorigenesis, highlighting its role as tumour suppressor.

### 4.1. Itch/Numb/GLI1 Axis

Numb is an evolutionary conserved developmental protein that plays crucial role in cell-fate determination and differentiation. Mammalian *Numb* encodes four alternatively spliced transcripts generating four proteins [56,57] that display several functions such as the control of asymmetric division, cell adhesion and migration, endocytosis and ubiquitylation of specific substrates [58,59]. The subversion of these Numb’s functions contributes to the upset of several malignancies, including cancer [60,61,62]. Numb shows tumour-suppressor function and its expression is frequently lost in lymphomas, breast tumours, non-small cell lung carcinomas (NSCLCs), salivary gland carcinoma and chronic myelogenous leukaemia (CML) [59].

Several evidences show that Numb determines cell-fate during nervous system development by promoting neuronal differentiation [57,63,64,65,66]. The asymmetric segregation of Numb between the daughter cells of neuronal precursors determines a double destiny: its expression correlates with the exit from cell cycle and the progression towards differentiation, while its lack is permissive for further proliferation. [57]. In particular, down-regulated expression of Numb in mouse cerebellum prevents the differentiation of GCPs, resulting in an enlarged external granule layer (EGL), where the cells are proliferating and in a deficit of internal granule layer (IGL) development, where mature granules reside [67]. However, the mechanisms by which Numb affects the neurogenesis are not yet deeply clarified. 

The structural properties of Numb make it an adaptor protein able to interact with various proteins, including E3 ubiquitin ligases such as Itch and Hdm2, by regulating both endocytosis and ubiquitylation of several substrates [59]. Indeed, Numb contains an amino-terminal phospho-tyrosine-binding domain (PTB) that mediates protein-protein interaction and a C-terminal proline rich region (PRR), containing putative Src homology 3-binding sites and Eps15 homology regions (DPF and NPF), involved in endocytosis events [56]. Through these functions, Numb controls cell signalling such as Notch and Hh pathways, crucial for cell-fate determination and maintenance of neural stem cell niches, respectively [58,59]. 

In particular, Numb directly interacts through its PTB domain with the WW domains of E3 ubiquitin ligase Itch, by promoting the ubiquitylation and degradation of Notch1, with consequent loss of Notch-dependent transcriptional activation [68].

Similarly, Numb has emerged as an Hh pathway suppressor that promotes GLI1 ubiquitylation [25] (Figure 3A). Since Numb does not have intrinsic ubiquitin-ligase activity, the ubiquitylation and degradation of GLI1 is consequent of the ability of Numb in recruiting the E3 ubiquitin ligase Itch on the substrate. Indeed, the three proteins, GLI1, Itch and Numb, form a complex and GLI1 ubiquitylation is strongly enhanced in presence of both Itch and Numb compared to the effect observed in the presence of only one of the two proteins. The direct interaction between Numb and Itch is necessary for the ubiquitylation of GLI1 (Figure 3A). Accordingly, Numb ΔPTB mutant, lacking the region responsible for the binding to Itch, fails to ubiquitylate GLI1. Likewise, the overexpression of Itch catalytic inactive mutant (C830A) prevents the Numb-induced GLI1 ubiquitylation, suggesting that Numb function requires the activity of Itch. 

It is known that Itch activity is regulated by conformational modifications; in particular, its catalytic activity is inhibited by the interaction of its C-terminal catalytic HECT domain to both a proline rich and WW motifs containing regions [49]. It has been shown that Numb binds the WW2 motif of Itch, making the WW region inaccessible for the binding with the HECT domain. In this manner, Numb destabilizes the intramolecular interaction of Itch, prevents its folding into an autoinhibitory state and induces its catalytic activity. Afterwards the activation of Itch, Numb allows GLI1 access and through direct interaction, mediated by Numb 174-421 aa region, recruits GLI1 in a complex with Itch. A direct binding between GLI1 and Itch is needed for the substrate ubiquitylation (Figure 3A). Generally, WW domains of HECT-E3 ubiquitin ligases interact with PPXY or pSP (proline residues preceded by phosphoserine) motifs of the substrate [69,70]. GLI1 contains two PPXY motifs in its C-terminal tails (region 755-1106) and a pS^1060^P motif that are all required for both recognition and subsequent ubiquitylation by Itch, by defining a novel Itch-dependent degron. Indeed, GLI1 triple mutant (GLI1^TM^) mutated in these three modules failed to bind Itch and is insensitive to Itch/Numb-mediated ubiquitylation, by determining increased proliferation, migration and invasion abilities, as well as transforming activity of medulloblastoma cells compared to wild type GLI1 [26]. 

These data confirm the role of Itch/Numb axis as negative regulator of Hh signalling and underline as alteration of this event could predispose to tumorigenesis. 

### 4.2. Itch/β-Arrestin2/SuFu Axis

β-arrestins are cytosolic multifunctional adaptor proteins identified as mediator of G protein-coupled receptors (GPCRs) signalling; after agonist stimulation, GPCRs are phosphorylated by GPCR kinases (GRKs) [71] thus promoting conformational changes that lead to activation of G proteins and results in the initiation of intracellular trafficking; this phosphorylation signal promotes β-arrestins recruitment [72]. β-arrestins were originally discovered as proteins able to desensitize G protein–mediated signalling but in the last decades they have emerged as versatile adaptor molecules involved in different physiological events, including the generation of protein systems in which arrestins act as scaffold proteins. The mammalian arrestin family includes four members: arrestin1 and arrestin4 are located into retinal rods and cones, respectively, while arrestin2 (also known as β-arrestin1) and arrestin3 (also known as β-arrestin2) are ubiquitously distributed and mediate GPCR desensitization and internalization [73]. β-arrestins emerged as important modulators of Hh signalling: in mammalian cells, the activation of the pathway leads to β-arrestin2 recruitment to the cell membrane where it promotes SMO endocytosis through a GRK-2-dependent phosphorylation [74]. Further, both β-arrestin1 and 2 are required for the formation of a “translocation complex” between SMO and a subunit of the kinesin-2 motor complex (Kif3A) that promotes the localization of SMO to primary cilia and the activation of downstream GLI transcription factors [75]. Moreover, β-arrestins may regulate Hh signalling through events that occur downstream of SMO; indeed, these proteins can act as general adaptors for different groups of E3s to promote ubiquitylation of specific substrates [76].

Within Hh pathway, it has been recently demonstrated that β-arrestin2 is required to promote Itch-mediated ubiquitylation of SuFu, the most important negative regulator of the Hh pathway [33]. The cytoplasmatic fraction of SuFu directly binds GLI proteins, thus regulating their transcriptional activity; in particular, when the Hh pathway is off, SuFu keeps GLI3 into the cytoplasm, protecting it from SPOP-mediated degradation and promoting its processing into a cleaved repressor form (GLI3R); signalling activation induces the dissociation of SuFu/GLI3 complex, causing GLI3 translocation into the nucleus where it is converted into a transcriptional activator [77]. The molecular mechanism that controls the formation of the SuFu/GLI3 complex has been recently clarified and it has been demonstrated how this mechanism has a protective role in MB oncogenesis. Specifically, SuFu directly interacts with WW1 and WW2 domains of E3 ubiquitin ligase Itch that in turn induces the poly-ubiquitylation of SuFu at lysines 321 and 457 [33]. This event, which does not target the protein for proteasomal degradation, involves the 63 lysine residue of the ubiquitin molecule [33] and it is known that K63-linked polyubiquitin chains are important for scaffold function of signalling proteins and positively regulate protein complex formation. Accordingly, SuFu K321/457R mutant, insensitive to Itch activity, shows a reduced ability to bind GLI3 leading to reduced synthesis of GLI3 repressor form. In this context, it has been observed that Itch-dependent SuFu poly-ubiquitylation is affected by the presence of β-arrestin2. In particular, β-arrestin2, SuFu and Itch form a trimeric complex in which the presence of β-arrestin2 is required to increase the association between Itch and SuFu [33] (Figure 3B). 

A recent whole genome sequencing of a large cohort of human Hh-driven MB (Hh-MBs) revealed new SuFu genetic mutations [78]; some of them occur in the C-terminal region of SuFu involving the 321 and 457 lysine residues and suggesting that disruptions in Itch-mediated SuFu poly-ubiquitylation is implicated in MB development. Indeed, in MB murine models it has been demonstrated that the expression of wild-type SuFu induces a reduction of the tumour volume, whereas SuFu K321/457R mutant does not [33]. Taken together these data demonstrate that the Itch-dependent poly-ubiquitylation of SuFu is strongly required for the negative regulation of Hh signalling and explain how alterations of this process, caused by SuFu mutations that makes it insensitive to Itch activity, contribute to MB oncogenesis (Figure 3B).

## 5. Itch and Its Role in Type I Non-Canonical Hh Pathway

In the last decade, several studies have shown that Hh signalling does not always culminate in GLI-mediated transcriptional activation and this subset of GLI-independent response is indicated as “non-canonical” Hh signalling [79]. It has been demonstrated that the regulation of PTCH1 receptor is required for the Hh canonical signalling pathway acting through the repression of SMO [80]. PTCH1 is also known to contain domains that suggest it may participate in the modulation of additional Hh-related molecular pathways, independently of its SMO-dependent functions [81,82,83,84,85]. Moreover, several evidences suggest that PTCH1 induces apoptosis independently from the canonical pathway [86], working as a dependence receptor when it is not occupied by the ligand [87,88]. For this process is essential the C-terminal domain (CTD) of PTCH1 [86], which contains a binding site for the scaffolding protein DRAL and TUCAN1 required for the recruitment of Caspase 9. The irreversible cell death is committed when Caspase 9, aggregated and activated by ubiquitylation, amplifies the proteolytic cascade, which culminates with the proteolysis of the PTCH1 CTD by Caspase 3 and Caspase 7 [88]. 

Recently, investigating on proteins that could modulate PTCH1 pro-apoptotic function, Chen and colleagues have found by mass spectrometry studies that the CTD of PTCH1 interacts with Itch and WWP2 ubiquitin E3 ligase [86] (Figure 4). At first, they have validated the mass spectrometry data by a co-immunoprecipitation assay, showing that Itch and WWP2 bind the CTD of PTCH1 in a stronger way than the other Nedd4-like family E3 ligase. Importantly, they demonstrated that Itch1 induces the strongest ubiquitylation of PTCH1 CTD when compared to the ubiquitylation levels obtained with other Nedd4 family members, including WWP2. Moreover, the authors showed that the endogenous depletion of Itch in HEK293T cells strongly reduced the full-length PTCH1 ubiquitylation level, characterizing Itch as the most promising E3 ubiquitin ligase candidate for PTCH1 in the absence of Hh ligands. Itch-mediated ubiquitylation leads to the degradation of PTCH1 and the catalytic activity of Itch is essential for this process, since the catalytically inactive mutant of Itch (C380A) does not induce any degradation of PTCH1. Moreover, WWP2 does not affect PTCH1 stability, consistently with the weaker ubiquitylation induced by this E3 ligase. The interaction between PTCH1 and Itch is dependent by the presence of two recognition motifs, PPPY and PPXY, contained in the CTD and in the cytoplasmic intracellular loop of PTCH1, respectively. Since the CTD of PTCH1 contains a single candidate ubiquitylation site (K1413 in human PTCH1), it has been investigated as the main target site of Itch (Figure 4). In this regard, the co-expression of the wild-type CTD with Itch induces a strong increase in the ubiquitylation of PTCH1 compared to endogenous levels, while PTCH1 ubiquitylation is absent in the K1413R mutant. Of note, the expression of Itch is able to induce a perinuclear localization of PTCH1 but not the mutant K1413R, indicating that the ubiquitylation at K1413 stimulates PTCH1 endocytosis. Finally, Itch overexpression is sufficient to increase GLI1 luciferase activity in NIH3T3 cells, indicating that PTCH1 degradation induced by Itch ubiquitylation leads to the lack of repression of SMO and to Hh pathway activation in the absence of the Hh ligand [86]. 

Nevertheless, although the Itch-mediated ubiquitylation at K1413 of PTCH1 is also found in response to the Hh stimuli, the absence of Itch does not prevent the degradation of PTCH1 by Hh, indicating that Itch is not able to regulate the canonical Hh pathway through the modulation of the PTCH1 basal turnover [86]. Indeed, Itch stands as a key regulator of the pro-apoptotic activity of PTCH1 in the absence of Hh in the Type I Non-Canonical Hh signalling [89], probably through a dual mechanism: the alteration of the pro-apoptotic complex, impairing the activation of Caspase 9 at the CTD of PTCH1 and the increase of internalization/degradation of PTCH1 (Figure 4). 

## 6. Concluding Remarks

Aberrant activation of Hh signalling is present in a wide spectrum of tumours, from solid to haematological and lymphoid malignancies, making Hh an attractive target for anticancer therapy. In the last decade, several Hh inhibitors and a various number of small molecules working as SMO or GLI antagonists have been designed and validated [90,91,92]. Although some of them have proven to be effective in Hh-dependent cancers treatment, the major issue in the employment of these compounds is the recurrence of target mutations or the presence of alternative mechanisms of activation. Consequently, multi-target therapy is emerging as a promising strategy approach for the treatment of Hh-dependent cancers. The best strategy envisioned so far is the development of further inhibitors able to target not only key components but also important regulators of the Hh pathway. 

Ubiquitylation processes are crucial mechanisms by which Hh pathway is controlled. Recently, the HECT E3 ligase Itch has emerged as an intriguing modulator of the Hh signalling. Indeed, it has been identified as a tumour suppressor that limits GLI1 oncogenic properties, [25,26] and enforces SuFu tumour suppressor functions, thus keeping the Hh pathway off [33]. Moreover, Itch was also found to regulate the basal turnover of PTCH1 in a context of Hh non-canonical pathway [86], thus linking this HECT E3 ligase to different aspects of Hh signalling. Further, given its key roles in immune regulation and the potential of the Hh pathway to modulate TCR signalling and adaptive immune response [93,94,95], Itch stands a promising therapeutic target in tumours arising from both solid and fluid organ system.

Itch has been shown to bind numerous substrates involved in controlling cell growth, differentiation and apoptotic processes and it is now clear that its deregulation can lead to malignant transformation and chemoresistance. As such, Itch and the proteins that regulate its function can represent exciting drug targets for the modulation of biological signalling and the identification of innovative therapeutic approaches. However, the potential for therapeutic targeting of Itch remains an underdeveloped area. In addition, while major efforts are dedicated to inhibit the activity of the E3 ligase enzymes, it would be also important thinking how to increase the Itch activity. In this regards, Lithium, a clinical mood stabilizer for the treatment of mental disorders, has been reported upregulate both mRNA and protein levels of Itch, thereby accelerating the degradation of GLI1 and leading to pancreatic cancer cell growth inhibition [42,96].

However, substantial gaps in our understanding of this important HECT E3 ligase will need to be filled. For example, additional structural information on the interaction between Itch and its substrates or its adaptor proteins or regulators might be valuable for drug design.

Interesting answers could arise from systematic studies on animal models aimed at correlating Itch ablation with tumorigenesis. Effects of a conditional Itchy mutation in a PTCH-MB prone background could be investigated in future studies. Itch mutant mice have a short life span due to severe immune and inflammatory disorders and spontaneous tumours have not yet been reported. This could be the result of redundancy with other Nedd4 family members and to the emerging number of Itch target proteins implicated in the Hh signalling network, in other developmental pathway and in cancer.

These and other potential questions will allow us to further explore the role Itch in Hh signalling and tumorigenesis and would provide a new road for drug intervention.

## Figures and Tables

**Figure 1 cells-08-00098-f001:**
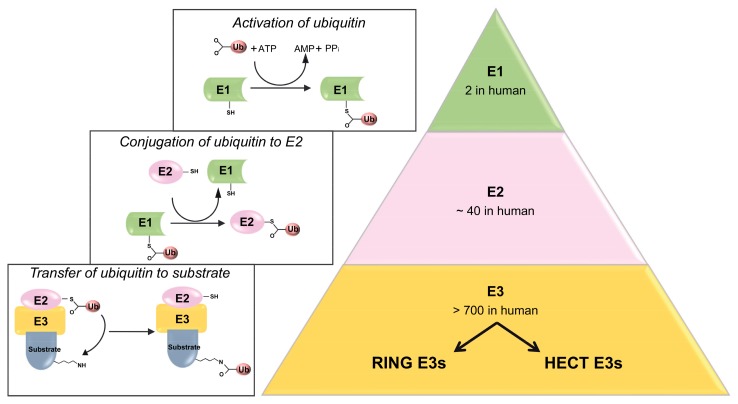
Ubiquitylation process. Ubiquitin (Ub) is attached to substrates by a cascade of reactions involving three enzymes. An E1 enzyme activates Ub in an ATP-dependent way. An E2 enzyme binds and then transfers the activated Ub to a substrate specifically bound to an E3 ligase. E3 enzyme can be divided into two main families: the RING and the HECT E3 ligases. Although, the ubiquitylation process usually leads to the degradation of the substrate, it can also drive regulative events.

**Figure 2 cells-08-00098-f002:**
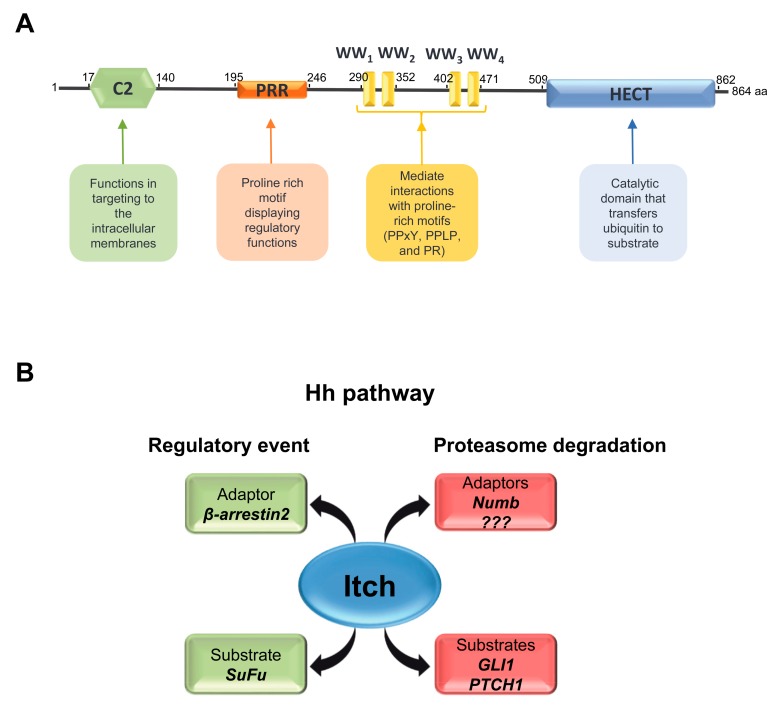
Schematic representation of the modular structure of Itch and its involvement in Hh pathway regulation. (**A**) Itch protein consists of an N-terminal Ca^2+^/lipid-binding (C2) domain (green hexagon), a proline rich motif (orange rectangle) displaying regulatory functions, a central region containing four WW domains (yellow bars) involved in protein-protein interaction and a catalytic HECT domain (blue rectangle). (**B**) The scheme shows the substrates of Itch involved in the regulation of Hh pathway (SuFu, GLI1 and PTCH1). Itch mediates regulatory events on SuFu and proteasome degradation on GLI1 and PTCH1 by the interaction with the adaptor proteins β-arrestin2 and Numb, respectively.

**Figure 3 cells-08-00098-f003:**
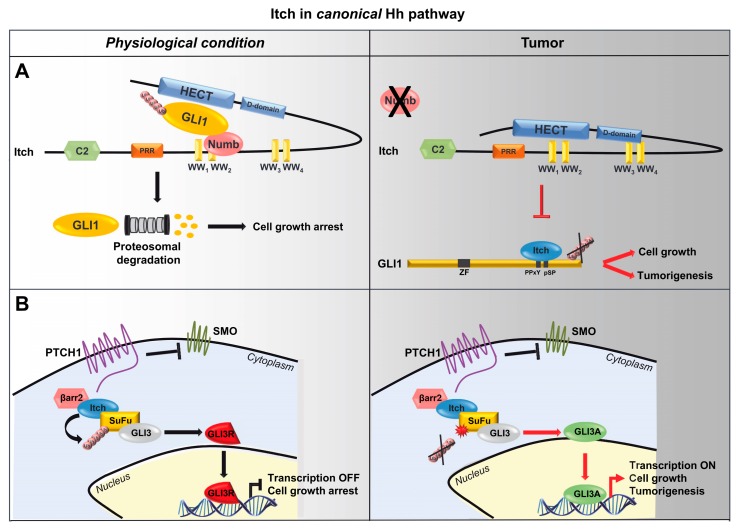
Role of Itch in canonical Hh pathway. (**A**) Model of the Numb/Itch-dependent regulation of GLI1 activity. Numb binds WW2 domain of Itch, releases Itch from its close-inactive conformation, thus allowing the recruitment of GLI1 and its degradation. In cancer, loss of Numb impairs this process enhancing GLI1 activity and promoting cell proliferation and tumorigenesis. (**B**) Model of the β-arrestin2/Itch-dependent regulation of the SuFu function. Itch, in complex with β-arrestin2, binds and ubiquitylates SuFu. This event does not lead to SuFu degradation but increases the association between SuFu and GLI3 promoting the generation of GLI3 repressor form (GLI3R) and inhibiting Hh signalling. Alterations in this mechanism, caused by SuFu mutations that make it insensitive to Itch-dependent ubiquitylation, are responsible for MB development.

**Figure 4 cells-08-00098-f004:**
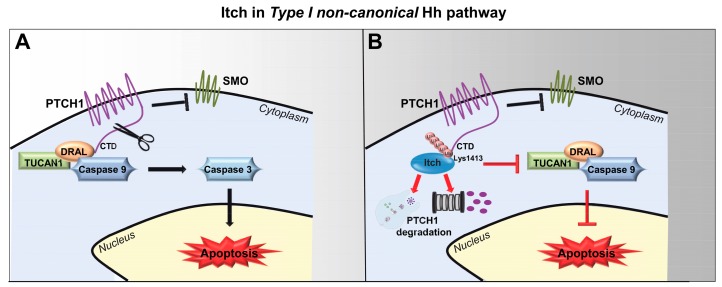
Role of Itch in Type I non-canonical Hh pathway. (**A**) In the absence of Hh, PTCH1 recruits a pro-apoptotic complex, including DRAL, TUCAN1 and Caspase 9, to its C-Terminal Domain (CTD). The activation of the Caspase 9 promotes the cleavage of the PTCH1 CTD by the Caspase 3, leading to irreversible commitment to cell death. (**B**) In the presence of Itch, the Lys1413 at the PTCH1 CTD is ubiquitylated, leading to the internalization and degradation of the PTCH1 and to the impairment of its pro-apoptotic activity.

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
