# Peer review of "Targeting Hedgehog Signalling through the Ubiquitylation Process: The Multiple Roles of the HECT-E3 Ligase Itch"

_cells, 2019, doi:10.3390/cells8020098_

Round 1

Reviewer 1 Report

A well written comprehensive review.   I recommend publication with minor modifications.  

Given that the review is focused on the way in which Itch regulates Hh signalling in the context of possible treatments for Hh-dependent cancers, it may also be relevant to mention studies showing that Hh signalling can modulate TCR signalling and the adaptive immune response (eg. Rowbotham Blood 2007; Furmanski JI 2013; Furmanski J Cell Science 2015).

Author Response

Review #1

Comments and Suggestions for Authors

Reviewer: A well-written comprehensive review. I recommend publication with minor modifications.  

Author: We thank the Reviewer for the positive comment on our manuscript.

Reviewer: Given that the review is focused on the way in which Itch regulates Hh signalling in the context of possible treatments for Hh-dependent cancers, it may also be relevant to mention studies showing that Hh signalling can modulate TCR signalling and the adaptive immune response (eg. Rowbotham Blood 2007; Furmanski JI 2013; Furmanski J Cell Science 2015).

Author: We have mentioned and discussed the role of Itch and Hedgehog pathway in modulating TCR signaling and immune response and added other references, as requested (pages 7, 17 and refs 93, 94, 95).

Reviewer 2 Report

 This is a broad review about role of Ubiquitination process in the regulation of the Sonic Hedgehog pathway with special emphasis on the HECT-E3 ligase Itch. Discussions of the overview of the Sonic Hedgehog pathways, introduction to the Ubiquitnation machinery and insight into the HECT E3 ligase Itch are thorough and clear. Role of HECT E3 ligase in the regulation of canonical Sonic Hedgehog pathway via the Itch/Numb/Gli1 axis and the beta-arrestin2/Itch/SuFu axis is entailed clearly. Also, the Type1 non-canonical regulation of Patch1 via Itch is described. The modulation of the Itch expression, understanding its interaction with other adaptor proteins to regulate the Sonic Hedgehog pathway, leading to the development of small molecule therapeutics for treatment of medulloblastoma serves the basis for discussion. 

Key critiques to consider: 

While reviewing the structure of Itch, it seems as a missed opportunity not to discuss the crystal structure of the Itch protein. The 3D structure of the protein has been elucidated (Wen et al, EMBO reports,2017) and also reveals the sites for the allosteric auto inhibition of Itch. This would help in appreciating the spatial orientation of the different domains of Itch and their movement in closed and open confirmations.

 Although the regulation of Itch has been mentioned, it needs to be elaborated in detail. 

The review in its current form needs to be thoroughly examined for spelling and grammatical errors. For example, the word type “type” is misspelled in line 305. 

Author Response

Review #2

Comments and Suggestions for Authors

Reviewer: This is a broad review about role of Ubiquitination process in the regulation of the Sonic Hedgehog pathway with special emphasis on the HECT-E3 ligase Itch. Discussions of the overview of the Sonic Hedgehog pathways, introduction to the Ubiquitination machinery and insight into the HECT E3 ligase Itch are thorough and clear. Role of HECT E3 ligase in the regulation of canonical Sonic Hedgehog pathway via the Itch/Numb/Gli1 axis and the beta-arrestin2/Itch/SuFu axis is entailed clearly. Also, the Type1 non-canonical regulation of Patch1 via Itch is described. The modulation of the Itch expression, understanding its interaction with other adaptor proteins to regulate the Sonic Hedgehog pathway, leading to the development of small molecule therapeutics for treatment of medulloblastoma serves the basis for discussion. 

Author: We thank the Reviewer for having appreciated the importance and soundness of our manuscript.

Reviewer: Key critiques to consider: 

While reviewing the structure of Itch, it seems as a missed opportunity not to discuss the crystal structure of the Itch protein. The 3D structure of the protein has been elucidated (Wen et al, EMBO reports, 2017) and also reveals the sites for the allosteric auto inhibition of Itch. This would help in appreciating the spatial orientation of the different domains of Itch and their movement in closed and open confirmations.

Author: We thank the reviewer to have raised this point. We have now discussed in more detail the structural organization of Itch including its crystallographic structure in a devoted issue (page 6 and refs 38, 40).

Reviewer: Although the regulation of Itch has been mentioned, it needs to be elaborated in detail. 

Author: We have implemented our manuscript including a paragraph on the regulation of Itch as suggested by the Referee to make the manuscript more complete (page 7 and refs 50, 52-55).

Reviewer: The review in its current form needs to be thoroughly examined for spelling and grammatical errors. For example, the word type “type” is misspelled in line 305. 

Authors: The revised manuscript has been re-edited in several parts.